# Therapeutic Vaccines for HPV-Associated Oropharyngeal and Cervical Cancer: The Next De-Intensification Strategy?

**DOI:** 10.3390/ijms23158395

**Published:** 2022-07-29

**Authors:** Grégoire B. Morand, Isabel Cardona, Sara Brito Silva Costa Cruz, Alex M. Mlynarek, Michael P. Hier, Moulay A. Alaoui-Jamali, Sabrina Daniela da Silva

**Affiliations:** 1Department of Otolaryngology Head and Neck Surgery, Sir Mortimer B. Davis-Jewish General Hospital, McGill University, Montreal, QC H3T 1E2, Canada; gregorie.morand@mcgill.ca (G.B.M.); sara.cruz@mcgill.ca (S.B.S.C.C.); alex.mlynarek@mcgill.ca (A.M.M.); michael.hier@mcgill.ca (M.P.H.); 2Department of Otolaryngology Head and Neck Surgery, Luzerner Kantonsspital, 6004 Lucerne, Switzerland; 3Department of Obstetrics and Gynecology, Montreal University Health Center, McGill University, Montreal, QC H3T 1E2, Canada; isabel.cardona@mcgill.ca; 4Segal Cancer Centre and Lady Davis Institute for Medical Research, Sir Mortimer B. Davis-Jewish General Hospital, Departments of Medicine and Oncology, Faculty of Medicine, McGill University, Montreal, QC H3T 1E2, Canada; moulay.alaoui-jamali@mcgill.ca; 5Human Immunology Research and Education, Federal University of Paraiba, João Pessoa 58059-900, PB, Brazil; 6Graduate Program in Dentistry, Federal University of Paraiba, João Pessoa 58059-900, PB, Brazil

**Keywords:** immune checkpoint inhibitors, radiotherapy, intensity-modulated, papillomavirus infections, head and neck neoplasms, immunization, vaccines

## Abstract

The rise in human papillomavirus (HPV)-associated oropharyngeal squamous cell carcinoma (OPSCC) has prompted a quest for further understanding of the role of high-risk HPV in tumor initiation and progression. Patients with HPV-positive OPSCC (HPV+ OPSCC) have better prognoses than their HPV-negative counterparts; however, current therapeutic strategies for HPV+ OPSCC are overly aggressive and leave patients with life-long sequalae and poor quality of life. This highlights a need for customized treatment. Several clinical trials of treatment de-intensification to reduce acute and late toxicity without compromising efficacy have been conducted. This article reviews the differences and similarities in the pathogenesis and progression of HPV-related OPSCC compared to cervical cancer, with emphasis on the role of prophylactic and therapeutic vaccines as a potential de-intensification treatment strategy. Overall, the future development of novel and effective therapeutic agents for HPV-associated head and neck tumors promises to meet the challenges posed by this growing epidemic.

## 1. Introduction

Oropharyngeal squamous cell carcinoma (OPSCC) arises most commonly in the palatine tonsils or the base of the tongue. As with other squamous cell carcinomas f the head and neck, it is classically caused by exposure to extrinsic carcinogens such as smoking and alcohol [1]. Since the late 1990s, there has been a rise in oropharyngeal cancer related to human papillomavirus (HPV) infection, especially in high-income countries [2,3]. On the other hand, the incidence of HPV-negative OPSCC has decreased by 50%, in accordance with the gradual reduction in tobacco and alcohol use since the 1980s [4]. HPV-related OPSCC shows a distinct epidemiology, clinical presentation and pathophysiology [5,6]. Patients are younger, healthier, and with a higher socioeconomic status [2,7]. They also have a better prognosis than their HPV-negative counterparts [1,8,9].

The discovery of the etiological role of HPV during OPSCC tumorigenesis is closely linked to research of cervical cancer of the uterus. The first reports pointing to the potential role of HPV in OPSCC were based on epidemiological studies demonstrating a correlation between the relative incidence of cancers of the cervix and cancers of the oropharynx in certain geographical areas [10,11]. Much of the research conducted into OPSCC has been based on the diagnostic and therapeutic approaches applied in cervical cancer. While some have proven to be successful, others have shown that OPSCC and cervical cancer are notably different [12,13]. This article reviews the differences and similarities in the pathogenesis and tumor progression of HPV-related OPSCC compared to. cervical cancer, with emphasis on the role of prophylactic and therapeutic vaccines as a potential de-intensified treatment strategy.

## 2. Epidemiology and Virus Transmission

Although cervical cancer and OPSCC have HPV as a common etiology, their respective tumors exhibit distinct epidemiological and transmission patterns (Table 1). First, virtually all cases of cervical cancer are provoked by HPV infection, while a proportion of OPSCC remains causally attributable to smoking and alcohol consumption [12]. These proportions vary greatly depending on the location where the studies were conducted. For instance, in North America over 80% of patients with OPSCC are HPV+, while in Italy, for example, only 20% of OPSCC patients are HPV+ [4,14].

It has been shown that high-risk HPV types are a major risk factor for cancer development. There are 15 high-risk HPV genotypes (HPV 16, 18, 31, 33, 35, 39, 45, 51, 52, 53, 56, 58, 59, 66 and 68). Among these 15 high risk HPV subtypes, the HPV16 and HPV18 genotypes cause approximately 70% of cervical cancers. The other subtypes are known to be carcinogenic in a lower proportion. For OPSCC, the HPV16 is the main subtype associated with more than 90% of the non-smoking-related cases [15] (Table 1).

HPV is transmitted sexually, and the spectrum of HPV-related cancers is associated—though not exclusively—with sexual practices, including vaginal and anal intercourse, hand-to-genital contact, oral sex, and/or the use of fomites (e.g., sex toys) [16]. The risk of HPV infection increases with the number of sexual partners [17] and an early age for first intercourse [18]. Men have a higher prevalence of HPV+ OPSCC than women and this difference is partially explained by their higher number of sexual partners overall. Vertical transmission from mother to child is rarely reported [16].

It has been demonstrated that, after inoculation of the virus, the HPV infection in the cervix resolves in more than 90% of cases after two years, while only 50% of women undergo seroconversion [19]. For HPV+ OPSCC, these numbers remain speculative at best since evidence is lacking. Absence of antibodies does not rule out the possibility that HPV infection has occurred, since HPV can be cleared without residual antibodies [12].

For cervical cancer, longitudinal studies have shown that cervical intraepithelial neoplasia (CIN) can occur after a few months [20]. Upon infection, the epithelial lesion can evolve from a low-grade squamous intraepithelial lesion (LSIL) to a high-grade squamous intraepithelial lesion (HSIL) to carcinoma in situ and then invasive malignancy. Gynecological examination with Papanicolaou screening has been successful at detecting early lesions and thereby preventing cervical cancer given that the progression from LSIL to cervical cancer is rather linear and is well understood (Figure 1). In other words, a progression to cervical cancer can be avoided if dysplasia is detected in a timely fashion and treated accordingly [20].

For OPSCC, the steps from infection to invasive cancer are not well understood. Further, since it is thought that infection occurs in the crypts of the palatine and base-of-tongue tonsils, the observation of the steps of carcinogenesis are rendered arduous (Figure 1). As we would expect, the screening of precursor lesions in the oropharynx lacks diagnostic accuracy [21]. Serological studies have shown that HPV+ OPSCC patients have antibodies against HPV more than 10 years before it becomes clinically apparent. It is, however, unclear when and how an individual with HPV infection will eventually develop OPSCC [22].

## 3. Structure and Oncogenesis of HPV

There are over 150 HPV strains (of which 15 are high-risk). HPV is a double-strand DNA virus measuring about 8000 base pairs [19]. Each genotype has a L1 nucleotide sequence that is at least 10% different from other types. The virus has a shape of an icosahedron with pentamers of L1 proteins building the shell of the virus. The variable part of the L1 genome corresponds to the outer surface of the pentamer, thus explaining the variable tropism of each HPV type [23]. HPV can affect the skin or the mucosa, but all high-risk oncogenic types are mucosotropic viruses. For example, HPV6 and HPV11 share 85% sequence identity and they both cause benign warts. HPV6 is more common in the anogenital region, while HPV11 is typically located in the upper airway, including the larynx. In contrast, HPV13, which shares 78% sequence identity with HPV6 and HPV11, does not cause any warts [24].

To infect the host cells, the virus binds to the basement membrane, which in turn activates the L2 protein (a minor capsid protein) via furin-mediated cleavage (Figure 2). This exposes the receptor-binding site of L1 [25] which allows for infection of basal layer keratinocytes [26]. After entering a basal layer keratinocyte, the virus maintains a low copy number, about 50 to 100 copies per cell. During this phase, plasmid or episomal maintenance of the viral genome is minimal [27]. When the infected basal cell layer keratinocytes stop multiplying and undergo differentiation and maturation into superficial layer keratinocytes, there is a massive upregulation of viral DNA replication and gene expression. The viral copy number is then thought to be at least 1000 copies per cell [28].

HPV DNA provides multiple open reading frames (ORF), designated as early (E) or late (L). E1 is a DNA replication enzyme essential for the amplification of the viral episome, while E2 is implicated in virus replication, transcription, and genome division during the viral life cycle. The structural proteins L1 and L2 are for the shell of the virus, as discussed above. The E3 ORF is lacking due to a sequencing error. The remaining E4 and E5 are involved in cell cycle entry, immune evasion and virus release [24]. HPV has only one DNA replication enzyme (E1) which makes viral replication mostly dependent upon cellular replication enzymes. Because these are only present in mitotically active cells, HPV viruses encode proteins that reactivate cellular DNA synthesis in mitotically inactive cells, inhibiting apoptosis and delaying the differentiation program of keratinocyte. The viral genes central to these functions are called E6 and E7 [28]. The E6 and E7 proteins are the major oncoproteins in high-risk HPV types, leading to deregulation of multiple signaling molecules, in particular by promoting degradation of the tumor suppressor p53 and retinoblastoma-associated (pRB) proteins, respectively (Figure 2) [29].

The HPV virus joins to the keratinocyte at the start of its journey and replicates in a cell that may die of natural causes; however, the virus does not induce necrosis per se, thus limiting inflammation and presentation of antigens to the immune system. It stays in the epithelium, thus avoiding a blood-borne or viremic phase [28]. An additional HPV-related immune escape mechanism is the inhibition of activated intraepithelial antigen-presenting cells, also known as Langerhans cells, when encountering HPV-like particles [30]. This explains why many HPV infections fails to induce seroconversions [31].

An HPV infection leading to cancer is the result of persistent infection and E6/E7 oncogene expression. This is the least common manifestation of HPV. Most infections are subclinical without any sequelae. Data indicate that viral DNA integration to the cell genome occurs frequently in cervical cancer, as episomal viral DNA can be successfully suppressed by IFNb-induced immune response [32]. Premalignant lesions seldom progress to invasive disease and will regress without treatment within two to five years in 60% to 80% of all cases. The more advanced the stage, the higher the risk of progression towards an invasive cancer [33].

## 4. Molecular Pathogenesis of HPV-Related Cancer

As discussed above, E6 and E7 proteins are crucial for HPV-driven carcinogenesis. It is known that HPVs exert their oncogenic role after DNA integration, gene expression of E5, E6 and E7 loci and p53/pRb host protein suppression, leading to increased cell proliferation and contributing to carcinogenesis. The E6 protein is a 150-amino-acid zinc-finger-bearing polypeptide. These zinc fingers are responsible for the enzymatic activity of E6. The primary interaction seen with E6 is to bind an alpha helical acidic LXXLL peptide [34]. Through this interaction, E6 proteins from HPV16 and HPV18 bind to the cellular tumor-suppressor P53, which ultimately induces the degradation of the protein. This occurs through the binding of the cellular E3 ubiquitin ligase E6AP to P53, resulting in its degradation by the 26S proteasome. Unlike high-risk HPV E6 proteins, low-risk mucosal and cutaneous E6 proteins are unable to induce degradation of the cellular P53 protein through the proteasome pathway [34,35]. The degradation results in an increase in tumor growth factors and a decrease in apoptosis. This leads to HPV replication not only in the lower levels of the epithelium in the basal layer, but also in the higher levels of the epithelium where the keratinocytes can differentiate into squamous cells [29]. However, E6 is also able to bind to P53 without inducing its degradation. This interaction prevents P53-related oncogenic functions such as transcriptional repression of TATA-containing promoters, since P53 bound to E6 cannot bind numerous P53-specific DNA recognition motifs. Within cervical tumors and in HPV-related oropharyngeal cancer, P53 is almost invariably wild type, which represents a major difference to most other solid cancers, including “classical” smoking-induced oropharyngeal cancer. Once cervical cancer tumors metastasize, mutations within P53 become more frequent [36]. The presence of mutant P53 may give such a cell a competitive advantage over cells in which P53 activity is merely abrogated by E6.

Aside from its main role in binding and degrading P53, a plethora of functions of the E6 proteins have been described [34]. E6s from high-risk mucosal HPVs and from certain cutaneous HPVs are capable of activating telomerase through a transcriptional up-regulation of the telomerase reverse transcriptase (TERT). A further difference between the E6 proteins of high-risk and low-risk HPV types is the induction of epithelial-to-mesenchymal transition; that is, NF-kappaB and Wnt activation, associated with miR regulation [34]. Finally, studies have shown that E6 is capable of downregulating specific genes that are involved in keratinocyte differentiation, by such means as NOTCH signaling by associating with the transcriptional coactivator MAML1 [35,37]. This explains the basaloid cell phenotype found in cervical cancer and HPV-positive oropharyngeal cancer, which is normally limited to the “basal” layer (or lower part) of the epithelium [38].

In differentiated keratinocytes, cellular DNA replication is switched off, meaning that high-risk HPV types have to stimulate replication in a more unnatural state. For this purpose, the role of E7 seems crucial to modulate the induction of DNA synthesis [39]. HPV E7 protein interacts with the so-called retinoblastoma protein pRb, which is a negative cell-cycle regulator involved in the G1/S and G2/M transitions. This occurs through E7 interaction with p21 which leads to its inhibition. Consequently, p21 cannot inhibit cyclin-dependent kinase, thus promoting G1/S and G2/M transition. One outcome of the inhibition of pRb by E7 is an increase in p16INK4A (cyclin-dependent kinase inhibitor 2A/multiple tumor suppressor 1) protein levels, which can be used as a surrogate marker for oncogenic HPV infection [40].

A further difference between high- and low-risk HPV types involves alternative splice variants. High-risk HPV transcription patterns involve a number of alternative splices which generate a complex pattern of mRNAs [41]. Although a few of the high-risk HPV genotypes are able to encode multiple E6 splice variants, all of the high-risk HPV genotypes encode the E6*I (hereinafter referred to as E6*) splice variant. E6* contains the first CXXC zinc-binding motif of E6. Although the exact functions of E6* remain controversial, clinical studies have shown that the frequency of the E6* mRNA isoform increases with lesion severity, being the most abundant viral transcript found in cervical cancer cells [42]. Recent studies indicate that E6* can promote viral genome integration into a host genome by inducing ROS and subsequently increasing genome instability. The NADPH oxidase pathway is thought to be the cellular HPV16 E6* leading to increased ROS production [43].

Interestingly, it seems that regulation of alternative splice variant E6* is dependent upon extracellular stimuli. In particular, E6* splicing is dependent on the presence of EGF and is mediated by the MAP kinase MEK1/2 pathway. On the other hand, it has been reported that E6* is able to inhibit the degradation of p53 by E6. Moreover, alternative splicing of E6 facilitates access to the E7 start codon and its translation. Hence, as a further consequence of E6 exon exclusion, higher levels of E7 protein are observed [44]. Taking these findings together, it has been postulated that EGF-dependent regulation of E6/E6* alternative splicing is an adaptive process of oncogene expression necessary for viral maturation [44]. Initially, HPV might require high levels of E6 to prevent apoptosis via p53 degradation, which would also favor chromosomal destabilization as an initial early event in carcinogenesis. With increasing differentiation, and decreasing EGF levels, cells start to splice their E6 to E6*, leading to increased p53 levels and high E7 expression and a consequent gain in selective growth advantage [44].

## 5. Prophylactic Vaccines

Prophylactic vaccines are directed against the L1 surface protein of the different HPV serotypes and can prevent primary infection of the cervix and tonsils. For cervical cancer, they lower the re-infection/recurrence rate after treatment for LSIL-HSIL. In contrast, therapeutic vaccines are directed against E6 and/or E7 proteins, expressed by HPV-induced invasive cancer of the cervix and oropharynx.

Experimental studies have shown that virus-like particles induce a strong response in both the adaptive and the innate immune systems, with robust generation of neutralizing antibodies [30,45]. The exact molecular mechanism used by prophylactic vaccine-induced antibodies to prevent infection is not fully understood but it seems that neutralizing antibodies bind to the virus’s L1 proteins prior to cell entry and thus prevent infection [28].

Today, several prophylactic HPV vaccines have been commercialized (Table 2). They are designed to induce an immune reaction with formation of neutralizing antibodies against the virus-like particles (VLPs) derived from HPV L1 capsid proteins. There are bivalent vaccines (HPV16, 18), one tetravalent vaccine (HPV6, 11, 16, 18), and the most recent vaccine is nonavalent (HPV6, 11, 16, 18, 31, 33,45, 52, 58) [28] (Table 2). Previous randomized controlled trials have shown vaccines to be highly efficacious with over 98% protection for HPV-related disease after a 5 to 6.8 years study period after vaccination [45]. Although L1 protein is specific for each HPV type, there is some cross-reactivity after HPV vaccination. A large protein such as L1 contains multiple epitopes, all of which induce an immune response. Some epitopes are immunodominant and induce a stronger response in the host, but the immune response of the host remains polyclonal. Usually, the immunodominant epitopes are designed to be type-specific to ensure an effective response against a particular HPV type; however, immune response to epitopes partially shared with other HPV types explain vaccine cross-reactivity [46].

Vaccination before puberty is now recommended in most Western countries [47]. It is expected that with adequate coverage of the adult population, there will be a reduced incidence of premalignant lesions of cervical cancer and OPSCC, respectively. However, since the latency period between the first inoculation and the development of cancer is so long, the effect of the vaccination implementation is not expected to be seen for another 20 years [47]. Therefore, it has been a matter of interest to evaluate the effect of vaccination in patients already suffering from HPV-related disease.

## 6. Use of Prophylactic Vaccines to Prevent Recurrent Disease

Several randomized controlled trials have been conducted in non-vaccinated women with cervical intraepithelial neoplasia (CIN), investigating the efficacy of vaccination in reducing the risk of recurrence/second primary tumor. In the SPERANZA project, women were randomly assigned to HPV vaccination after having being treated for CIN 2 with a loop electrosurgical excision procedure [50]. The recurrence rate was significantly lower in the vaccination group (risk reduction of 81.2%). In a recent meta-analysis of six studies of HPV-unvaccinated patients with CIN 2+, the risk of recurrence was lower among patients who underwent excision followed by adjuvant HPV vaccination compared with excision alone (1.9 vs. 5.9%, relative risk 0.36, 95% CI 0.23–0.55) [51].

Since HPV vaccines prevent virus inoculation by producing antibodies against L1 protein and given the fact that L1 is not expressed during the oncogenic process, it is thought that the reduction in recurrence risk is due either to prevention of new inoculation (possibly by another strain) or to auto-inoculation from episodic dormant viral activation. In other words, vaccines are mostly efficacious since they prevent a new (at least to the immune system) HPV strain from infecting the host. Therefore, one may argue that vaccines do not prevent recurrence, but do prevent second primary tumors [51].

For OPSCC, there are no data available on the effect of prophylactic vaccines in patients with HPV-related disease. New inoculation or auto-inoculation is also possible in cancer of the oropharynx. However, the progression from inoculation to cancer development cannot be determined in OPSCC and the latency period is thought to be as long as 20 years, making studies much harder to be carried out or reproduced [22]. On the other hand, epidemiological studies have shown that the risk of HPV-related second primary cancers is significantly higher among HPV associated cancer survivors, thus providing a logically consistent argument for secondary vaccination to help reduce any risk of re-inoculation [52], even if it may be of limited use for the oropharynx only.

## 7. Therapeutic Vaccines

Therapeutic vaccines differ from prophylactic vaccines, as they are designed to generate cell-mediated immunity against transformed cells [53], rather than neutralize antibodies. Among the HPV proteins, the E6 and E7 oncoproteins are considered almost ideal as targets for immunotherapy treatment of cervical and oropharyngeal cancer, as they are essential for the onset and evolution of malignancy and are constitutively expressed in both premalignant and invasive lesions [54]. Therapeutic vaccine design studies have shown that constructs of both E6 and E7 antigens are more efficacious at generating strong immune response than vaccines targeting a single antigen (E6 or E7) [53]. Furthermore, vaccines administered by electroporation induced the strongest immune response [53]. Therapeutic vaccines have also been shown to induce potent and durable immune response in Phase 1 clinical studies [55].

Another viable strategy is to use RNA vaccines to induce immune response against the E6 and E7 antigens [56]. The intravenously administered cancer mRNA vaccine efficiently primes and expands antigen-specific effector and memory CD8+ T cells [56]. Further, it was shown that mRNA vaccines could sensitize refractory tumors to a checkpoint blockade.

In recurrent/metastatic therapeutic settings, treatment with check-point inhibitors has become the standard of care for oropharyngeal cancer since the results of the Keynote-048 trial were published [57]. However, to build on this advance, there is now great interest in the additional use of therapeutic vaccines to further improve standards of care.

There is limited evidence from Phase II trials indicating positive results in patients with recurrent/metastatic HPV-related disease who received checkpoint inhibitors (PD1/PDL1) in combination with E6/E7-therapeutic vaccines [58,59] (Table 3). These studies show higher overall response rates when compared to historical cohorts in patients with recurrent/metastatic HPV-related cancer treated with checkpoint inhibitors only. Such studies suggest that the strength of the immune response induced by the therapeutic HPV vaccines correlates with overall patient survival rates [60].

Several clinical trials with E6/E7 vaccines are in progress in the recurrent/metastatic setting (NCT03260023, NCT04180215, cervISA/*NCT02128126,* NCT04405349) for HPV-positive disease. The combination of E6-E7 immunization and inhibition of the PD1-PDL1 checkpoint, which is a mechanism of immune evasion in advanced cancers of the head and neck, is expected to double the natural immune system response against the cancer [61].

Concerning definitive/curative settings, radiation with high-dose cisplatin remains the standard of care in HPV+ OPSCC, since two randomized controlled trials (DeEscalate and RTOG 1016) have demonstrated that standard therapy with concomitant high-dose cisplatin effects a better cure rate than the less toxic cetuximab [62,63]. Current studies are evaluating the possibility of replacing cisplatin with checkpoint inhibitors as a de-intensification strategy (e.g., HN5 [64] and REACH [65]).

Similarly, in the NCT04369937 trial, intermediate-risk patients with HPV-related disease will undergo intensity modulated radiotherapy (IMRT) combined with Cisplatin, Pembrolizumab, and an experimental E6–E7 vaccine. The combination of checkpoint inhibitors with vaccines is of great interest, especially after the negative results of the JAVELIN 100 study, which showed no survival advantage in patients receiving Avelumab (compared to a placebo) in addition to radiation with high-dose cisplatin [66,67]. However, trials similar to JAVELIN 100 in lung and esophageal cancer showed that the addition of a checkpoint inhibitor in an adjuvant rather than a concomitant setting did lead to an improvement in survival rates, suggesting that the timing of the immune system stimulation is critical in improving of radiation therapy-related outcomes [66]. Interestingly, treatment with checkpoint inhibitors before radiation enhanced the local tumor response but abrogated the abscopal effect. On the other hand, treatment with checkpoint inhibitors after radiotherapy induced a greater abscopal effect and were associated with the best survival rates [68], although the local tumor response was not as strong as when checkpoint inhibitors were given before radiotherapy.

Another de-intensification strategy is to begin treatment with neoadjuvant chemotherapy or immunotherapy to reduce the tumor load, such as in the NECTORS trial [69]. Some trials (NCT05232851) are also investigating the possibility of adding therapeutic vaccines to checkpoint inhibitors in the neoadjuvant setting. The rationale here is to reduce the size of the tumor before surgery and thus reduce the amount of normal tissue that needs to be removed, thus de-intensifying the treatment [6].

## 8. Conclusions

There are numerous emerging prospects for the beneficial use of therapeutic vaccines, as well as for targeted, molecular-based therapies for HPV+ OPSCC. Overall, the future for developing novel and effective therapeutic agents for HPV-associated head and neck tumors is promising and continued progress is critical to meet medical challenges posed by this growing epidemic.

## Figures and Tables

**Figure 1 ijms-23-08395-f001:**
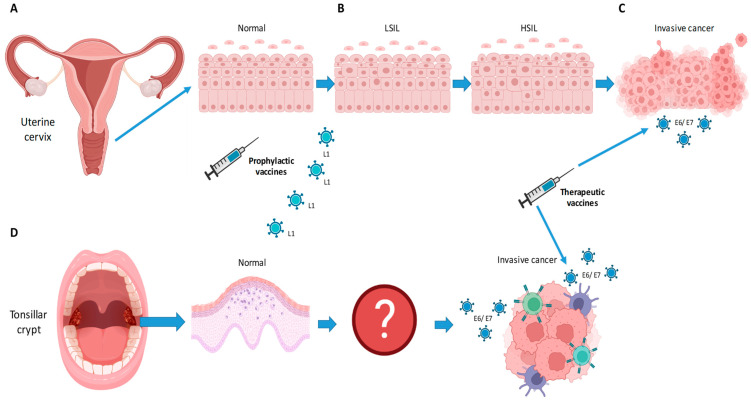
Schematic drawing in (**A**–**C**) illustrates the progression from normal epithelium to low-grade squamous intraepithelial lesion (LSIL), high grade squamous intraepithelial lesion (HSIL), and invasive cancer, upon infection with HPV virus in cervical cancer. For oropharyngeal cancer (**D**), the virus is thought to infect the epithelium of the tonsillar crypt. The steps leading to invasive cancer remain unknown. Scheme created by IC. Image was created using Biorender Free Software.

**Figure 2 ijms-23-08395-f002:**
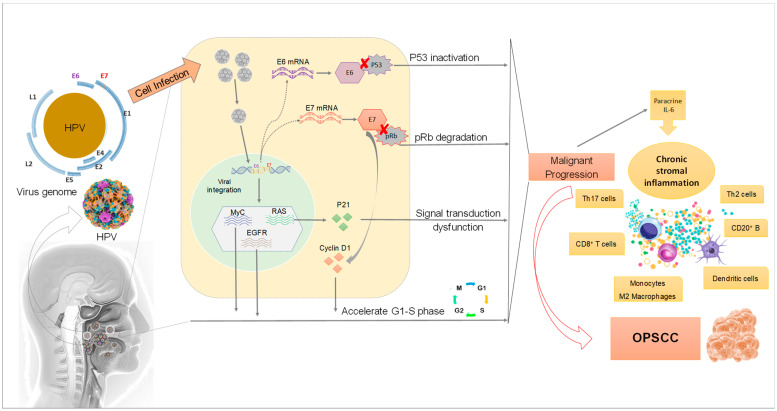
Mechanisms of HPV infection affecting the development of OPSCC. Once HPV infects epithelium cells in the head and neck region, its DNA is integrated into the host cell genome resulting in progressive alterations in proto-oncogenes and tumor suppressor genes. The overexpression of E6 and E7 associated with protein dysregulation in the host cells leads to dysfunction in cell metabolism and malignant proliferation.

**Table 1 ijms-23-08395-t001:** Comparison of cervical and oropharyngeal HPV+ cancer.

Characteristics	Cervical Cancer	Oropharyngeal Cancer
Etiology	HPV > 99%	HPV, smoking, alcohol
Developing/developed country	9:1	1:3
Trends	Decreasing in developed countries Increasing in developing countries	Increasing
HPV16	61% of all cases	>90% of HPV-related cases
HPV18	10% of all cases	2% of HPV-related cases
E6/E7	Present	Present
Rb/p53	Wt	Wt
Transmission	Sexual	Sexual (incl. French kissing)
Latency	Shortto CIN	>10 years (to SCC)
Natural history	CIN1, CIN2, CIN3, CIS, invasive SCC (median age 49 y)	Unknown (median age 54 y)
Screening	Yes, Pap smear	“Oro-pap“ smear does not work
Effect of “primary“ vaccination?	Yes (20 years)	Yes (20 years)
Effect of “secondary“ vaccination?	Yes (CIN 2+)	Unknown
Five-year survival rates	68%	95%

**Table 2 ijms-23-08395-t002:** The most recent HPV prophylactic vaccines recommended by the Centers for Disease Control and Prevention (CDC) Advisory Committee on Immunization Practices (ACIP) *.

Vaccine Name	9vHPV	4vHPV	2vHPV
Brand name	Gardasil 9, Merck	Gardasil, Merck	Cervarix, GlaxoSmithKline
Vaccine type	Nonavalent	Quadrivalent	Bivalent
Serotypes covered	HPV 6, 11, 16, 18, 31, 33, 45, 52, 58	HPV 6, 11, 16, 18	HPV 16, 18
Manufacturing	*Saccharomyces cerevisiae* (Baker’s yeast)—expressing L1	*Saccharomyces cerevisiae* (Baker’s yeast)—expressing L1	*Trichoplusia ni* insect cell line infected with L1 encoding recombinant baculovirus
Adjuvant	500 µg amorphous aluminum hydroxyphosphate sulfate	225 µg amorphous aluminum hydroxyphosphate sulfate	500 µg aluminum hydroxide 50 µg 3-O-desacyl-4′ monophosphoryl lipid A
Available on market (USA)	2014–present	2006–2016	2009–2016
Target age	11–12 years old, licensed for 9–26 years old (up to 45 years old)		
Target group	Females since 2006, males since 2011		
Schedule	For persons < 15 years-old, two doses (0 and 6–12 months) For persons > 15 years-old, three doses (0, 1–2, and 6 months)		
Administration	Intramuscularly (deltoid muscle), 0.5 mL vial, storage 2–8 °C		
Side effects	Local: injection-site-related pain, swelling, and erythema Systemic: headache, dizziness, myalgia, arthralgia, and gastrointestinal symptoms No evidence for Guillain-Barré Syndrome, complex regional pain syndrome (CRPS) and/or postural orthostatic tachycardia syndrome (POTS) [48].		
Contraindications	Hypersensitivity to yeast	Hypersensitivity to yeast	Anaphylactic latex allergy

* Table was created using information provided by Meites E, Szilagyi PG, Chesson HW, Unger ER, Romero JR, Markowitz LE. Human Papillomavirus Vaccination for Adults: Updated Recommendations of the Advisory Committee on Immunization Practices. MMWR Morb Mortal Wkly Rep 2019;68:698–702 [49].

**Table 3 ijms-23-08395-t003:** Overview of therapeutic vaccines for HPV-positive oropharyngeal and cervical cancer.

Therapeutic Setting	Trial Name	Country	Institution	PI	Population	Intervention	Comparison	Outcome
Definitive/Curative Setting	NCT05232851	USA	Mayo	David M Routman	24 patients with locally advanced HPV-positive oropharyngeal cancer	Neoadjuvant Pembrolizumab + Liposomal HPV-16 E6/E7 Multipeptide Vaccine PDS0101	Neoadjuvant Pembrolizumab alone	PFS and OS
	NCT02405221	USA	Johns Hopkins	Stéphanie Gaillard	14 patients with history of HPV positive cervical cancer	Adjuvant L2E6E7 vaccination (TA-CIN)	Single arm open label	Recurrence rate
	NCT04369937	USA	Pittsburgh	Dan Zandberg	50 patients with intermediate-risk HPV positive oropharyngeal cancer	Radiation + Cisplatin + Pembrolizumab + E6/E7 vaccination (ISA 101)	Single arm open label	PFS
Recurrent/Metastatic Setting	NCT02426892 [58]	USA	MD Anderson	Massarelli	34 patients with incurable HPV+ solid tumors	Nivolumab + HPV E6/7 vaccination (ISA 101)	Single arm open label	Overall response rate
	NCT03444376 [59]	South Korea	Pohang	Soo-Young Hur	60 patients with advanced HPV16 or HPV18-positive cervical cancer	Pembrolizumab + HPV E6/E7 vaccination (GX-188E)	Single arm open label	Overall response rate
	CerviISA NCT02128126	Belgium/Germany/Netherlands	Multicentric	Winald Gerritsen	93 patients with advanced HPV16-positive cervical cancer	Carboplatin and Paclitaxel with or without Bevacizumab + HPV E6/7 vaccination (ISA 101)	Single arm open label	Overall response rate
	NCT04405349	Central Europe	Multicentric	Nykode Therapeutics	50 patients with unresectable HPV-positive cervical cancer.	Atezolizumab + HPV E6/7 vaccination (VB10.16)	Single arm open label	Overall response rate
	NCT03260023	USA/France/Spain	Multicenter	Transgene	150 patients with HPV-positive unresectable malignancies	Avelumab + HPV E6/7 vaccination (TG4001)	Avelumab alone	Overall response rate
	NCT04180215	USA	Multicenter	Hookipa Biotech	200 patients with HPV16-positive cancer, unresectable	HPV E6/7 vaccination, i.v. and intratumoral	Single arm, open label	Overall response rate

## Data Availability

The datasets generated for this study can be obtained upon reasonable request by email to the corresponding author.

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
