# Peer review of "Therapeutic Vaccines for HPV-Associated Oropharyngeal and Cervical Cancer: The Next De-Intensification Strategy?"

_ijms, 2022, doi:10.3390/ijms23158395_

Round 1

Reviewer 1 Report

The aim of this review was to outline a number of novel therapeutic vaccine strategies for the potential treatment and de-intensification of individuals with HPV+ OPSCC. Although this review contained many great things, I feel like a few things can improve this manuscript prior to publication, and therefore recommend Major revisions prior to acceptance.

ISSUES:

1. English grammar and spelling issues throughout manuscript

2. If possible, could Figure 1 be of better quality, it was difficult to see much of the detail.

3. In table 2, under the Manufacturing row, you have L1 with a superscript, not sure what that is supposed to mean, but there is no information in the footnotes, so that needs to be addressed.

4. I think the biggest issue, is that based on the title, I expected a comprehensive look at therapeutic vaccines, but instead I get a brief overview in the form of the penultimate section 7. I think more needs to be added, and can be added. 

5. Also, if you do add more, per the suggestion in 4, then a table would be nice to summarize all that information. One that includes trial name, status, relevant pubs, protein targets, etc.

Author Response

We thank the reviewer for the thorough review of our manuscript and the valuable comments, which we have addressed and have strengthened the quality of our work. Please see below our point-by-point responses and the corrections we made in the revised manuscript (using track-changes for your convenience).

The aim of this review was to outline a number of novel therapeutic vaccine strategies for the potential treatment and de-intensification of individuals with HPV+ OPSCC. Although this review contained many great things, I feel like a few things can improve this manuscript prior to publication.

  1. English grammar and spelling issues throughout manuscript

Response: English grammar was checked.

  1. If possible, could Figure 1 be of better quality, it was difficult to see much of the detail.

Response: Figure 1 was modified to allow better resolution/visualization.

  1. In table 2, under the Manufacturing row, you have L1 with a superscript, not sure what that is supposed to mean, but there is no information in the footnotes, so that needs to be addressed.

Response: We apologize for this inadvertent typo. No superscript is supposed to follow L1. Those were erased.

  1. I think the biggest issue, is that based on the title, I expected a comprehensive look at therapeutic vaccines, but instead I get a brief overview in the form of the penultimate section 7. I think more needs to be added, and can be added. 

Response: We appreciated these comments. Section 7 was revised and expanded. A new table (Table 3) was added with an overview of therapeutic vaccines for HPV-positive oropharyngeal and cervical cancer.

  1. Also, if you do add more, per the suggestion in 4, then a table would be nice to summarize all that information. One that includes trial name, status, relevant pubs, protein targets, etc.

Response: A new table (Table 3) was added in the manuscript as your suggestion.

Thank you.

Table 3: Overview of therapeutic vaccines for HPV-positive oropharyngeal and cervical cancer

Therapeutic setting

Trial name

Country

Institution

PI

Population

Intervention

Comparison

Outcome

Definitive/Curative Setting

NCT05232851

USA

Mayo

David M Routman

24 locally advanced HPV positive oropharyngeal cancer

Neoadjuvant Pembrolizumab + Liposomal HPV-16 E6/E7 Multipeptide Vaccine PDS0101

Neoadjuvant Pembrolizumab alone

PFS and OS

NCT02405221

USA

Johns Hopkins

Stéphanie Gaillard

14 patients with history of HPV positive cervical cancer

Adjuvant L2E6E7 vaccination (TA CIN)

Single arm open label

Recurrence rate

NCT04369937

USA

Pittsburgh

Dan Zandberg

50 patients with intermediate risk HPV positive oropharyngeal cancer

Radiation + Cisplatin + Pembrolizumab + E6/E7 vaccine (ISA 101)

Single arm open label

PFS

Recurrent/Metastatic Setting

NCT02426892[58]

USA

MD Anderson

Massarelli

34 patients with incurable HPV+ solid tumors

Nivolumab + HPV E6/7 vaccination (ISA 101)

Single arm open label

Overall response rate

 NCT03444376[59]

South Koread

Pohang

Soo-Young Hur

60 patients with advanced HPV16 or HPV18 positive cervical cancer

Pembrolizumab + HPV E6/E7 vaccination (GX-188E)

Single arm open label

Overall response rate

CerviISA

NCT02128126

Belgium/Germany/Netherlands

Mulitcentric

Winald Gerritsen

93 patients with advanced HPV16 positive cervical cancer

Carboplatin and Paclitaxel with or without Bevacizumab + HPV E6/7 vaccination (ISA 101

Single arm open label

Overall response rate

NCT04405349

Central Europe

Multicentric

Nykode Therapeutics

50 patients with unresectable HPV positive cervical cancer.

Atezolizumab + HPV E6/7 vaccine (VB10.16)

Single arm open label

Overall response rate

NCT03260023

USA/France/Spain

Multicenter

Transgene

150 patients with HPV positive unresectable malignancies

Avelumab + HPV E6/7 vaccines (TG4001)

Avelumab alone

Overall response rate

NCT04180215

USA

Multicenter

Hookipa Biotech

200 HPV16 positive cancer, unresectable

HPV E6/ vaccine

Single arm, open lable

Overall response rate

Reviewer 2 Report

This is an interesting review about vaccines for HPV-associated oropharyngeal cancer.

The paper is well written. However, some issues remain.

Since the article reviews the differences and similitudes in the pathogenesis and tumor progression of HPV-related OPSCC vs. cervical cancer, the title and the abstract must report it.

More data about studies on therapeutic vaccines, such as survival percentages and side effects in different arms, should be added.

Author Response

We thank the reviewer for the thorough review of our manuscript and the valuable comments, which we have addressed and have strengthened the quality of our work. Please see below our point-by-point responses and the corrections we made in the revised manuscript (using track-changes for your convenience).

This is an interesting review about vaccines for HPV-associated oropharyngeal cancer.

The paper is well written. However, some issues remain.

  1. Since the article reviews the differences and similitudes in the pathogenesis and tumor progression of HPV-related OPSCC vs. cervical cancer, the title and the abstract must report it.

Response: The title and abstract were amended accordantly.

  1. More data about studies on therapeutic vaccines, such as survival percentages and side effects in different arms, should be added.

Response: Section 7 was revised and expanded as requested. A new table (Table 3, please see below) was added with an overview of therapeutic vaccines for HPV-positive oropharyngeal and cervical cancer as you suggested. Thank you.

Table 3: Overview of therapeutic vaccines for HPV-positive oropharyngeal and cervical cancer

Therapeutic setting

Trial name

Country

Institution

PI

Population

Intervention

Comparison

Outcome

Definitive/Curative Setting

NCT05232851

USA

Mayo

David M Routman

24 locally advanced HPV positive oropharyngeal cancer

Neoadjuvant Pembrolizumab + Liposomal HPV-16 E6/E7 Multipeptide Vaccine PDS0101

Neoadjuvant Pembrolizumab alone

PFS and OS

NCT02405221

USA

Johns Hopkins

Stéphanie Gaillard

14 patients with history of HPV positive cervical cancer

Adjuvant L2E6E7 vaccination (TA CIN)

Single arm open label

Recurrence rate

NCT04369937

USA

Pittsburgh

Dan Zandberg

50 patients with intermediate risk HPV positive oropharyngeal cancer

Radiation + Cisplatin + Pembrolizumab + E6/E7 vaccine (ISA 101)

Single arm open label

PFS

Recurrent/Metastatic Setting

NCT02426892[58]

USA

MD Anderson

Massarelli

34 patients with incurable HPV+ solid tumors

Nivolumab + HPV E6/7 vaccination (ISA 101)

Single arm open label

Overall response rate

 NCT03444376[59]

South Koread

Pohang

Soo-Young Hur

60 patients with advanced HPV16 or HPV18 positive cervical cancer

Pembrolizumab + HPV E6/E7 vaccination (GX-188E)

Single arm open label

Overall response rate

CerviISA

NCT02128126

Belgium/Germany/Netherlands

Mulitcentric

Winald Gerritsen

93 patients with advanced HPV16 positive cervical cancer

Carboplatin and Paclitaxel with or without Bevacizumab + HPV E6/7 vaccination (ISA 101

Single arm open label

Overall response rate

NCT04405349

Central Europe

Multicentric

Nykode Therapeutics

50 patients with unresectable HPV positive cervical cancer.

Atezolizumab + HPV E6/7 vaccine (VB10.16)

Single arm open label

Overall response rate

NCT03260023

USA/France/Spain

Multicenter

Transgene

150 patients with HPV positive unresectable malignancies

Avelumab + HPV E6/7 vaccines (TG4001)

Avelumab alone

Overall response rate

NCT04180215

USA

Multicenter

Hookipa Biotech

200 HPV16 positive cancer, unresectable

HPV E6/ vaccine

Single arm, open lable

Overall response rate

Reviewer 3 Report

The submitted review article outlines a number of emerging prospects for therapeutic vaccines for HPV+ OPSCC and considers this as a novel deintensification strategy. The topic is interesting and clinically highly relevant. 

Comments

The article has a strong background on HPV, cervix and oropharynx HPV infection and also on prophylactic vaccines, which is valuable, but the main point would be the treatment deintensification in OPSCC due to therapeutic vaccines, which comes too short in the manuscript. 

The article mentions combinations of  E6/E7-therapeutic vaccines with PD1/PDL1 checkpoint inhibitor therapy, which is not a standard treatment of OPSCC. Also the deintensification or de-escalation for a head and neck oncologist would mean dose reduction of cisplatin or radiation, which is not discussed in the review. 

Please include a section on standard treatment of OPSCC, the state and sense of checkpoint inhibitors in relation to standard treatment, in HPV+ OPSCC; how, the E6/E7 therapeutic vaccinnation is expected to be performed in the treatment schedule as first line treatment and adjuvant therapy? OPSCC can be treated as surgical first line, followed by PORT, or with radiochemotherapy as first line. HPV levels can be followed by circulating tumor DNA measurements or similar. Studies are available that intend to follow up the HPV+ OPSCC treatment schedule and define treatment times, also for E6/E7 vaccines. The only discussion about immune checkpoint regulators targeted therapy is not comparable with the promised de-escalation strategy. 

In 2022 RNA vaccines should be also mentioned and discussed. Please find a possible source for this point in a recent article:

Oncoimmunology. 2019; 8(9): e1629259., DOI: 10.1080/2162402X.2019.1629259

As mentioned in the manuscript, the response rate in OPSCC to standard therapy is relative high. Nevertheless, very few cases do not benefit from the therapy, which require alternative treatments. Do these cases have a chance by the therapeutic vaccines treatments?

Author Response

We thank the reviewer for the thorough review of our manuscript and the valuable comments, which we have addressed and have strengthened the quality of our work. Please see below our point-by-point responses and the corrections we made in the revised manuscript (using track-changes for your convenience).

The submitted review article outlines a number of emerging prospects for therapeutic vaccines for HPV+ OPSCC and considers this as a novel deintensification strategy. The topic is interesting and clinically highly relevant.

  1. The article has a strong background on HPV, cervix and oropharynx HPV infection and also on prophylactic vaccines, which is valuable, but the main point would be the treatment deintensification in OPSCC due to therapeutic vaccines, which comes too short in the manuscript.

Response: Section 7 was revised and expanded. A new table (Table 3, please see below) was added with an overview of therapeutic vaccines for HPV-positive oropharyngeal and cervical cancer.

Table 3: Overview of therapeutic vaccines for HPV-positive oropharyngeal and cervical cancer

Therapeutic setting

Trial name

Country

Institution

PI

Population

Intervention

Comparison

Outcome

Definitive/Curative Setting

NCT05232851

USA

Mayo

David M Routman

24 locally advanced HPV positive oropharyngeal cancer

Neoadjuvant Pembrolizumab + Liposomal HPV-16 E6/E7 Multipeptide Vaccine PDS0101

Neoadjuvant Pembrolizumab alone

PFS and OS

NCT02405221

USA

Johns Hopkins

Stéphanie Gaillard

14 patients with history of HPV positive cervical cancer

Adjuvant L2E6E7 vaccination (TA CIN)

Single arm open label

Recurrence rate

NCT04369937

USA

Pittsburgh

Dan Zandberg

50 patients with intermediate risk HPV positive oropharyngeal cancer

Radiation + Cisplatin + Pembrolizumab + E6/E7 vaccine (ISA 101)

Single arm open label

PFS

Recurrent/Metastatic Setting

NCT02426892[58]

USA

MD Anderson

Massarelli

34 patients with incurable HPV+ solid tumors

Nivolumab + HPV E6/7 vaccination (ISA 101)

Single arm open label

Overall response rate

 NCT03444376[59]

South Koread

Pohang

Soo-Young Hur

60 patients with advanced HPV16 or HPV18 positive cervical cancer

Pembrolizumab + HPV E6/E7 vaccination (GX-188E)

Single arm open label

Overall response rate

CerviISA

NCT02128126

Belgium/Germany/Netherlands

Mulitcentric

Winald Gerritsen

93 patients with advanced HPV16 positive cervical cancer

Carboplatin and Paclitaxel with or without Bevacizumab + HPV E6/7 vaccination (ISA 101

Single arm open label

Overall response rate

NCT04405349

Central Europe

Multicentric

Nykode Therapeutics

50 patients with unresectable HPV positive cervical cancer.

Atezolizumab + HPV E6/7 vaccine (VB10.16)

Single arm open label

Overall response rate

NCT03260023

USA/France/Spain

Multicenter

Transgene

150 patients with HPV positive unresectable malignancies

Avelumab + HPV E6/7 vaccines (TG4001)

Avelumab alone

Overall response rate

NCT04180215

USA

Multicenter

Hookipa Biotech

200 HPV16 positive cancer, unresectable

HPV E6/ vaccine

Single arm, open lable

Overall response rate

  1. The article mentions combinations of E6/E7-therapeutic vaccines with PD1/PDL1 checkpoint inhibitor therapy, which is not a standard treatment of OPSCC. Also the deintensification or de-escalation for a head and neck oncologist would mean dose reduction of cisplatin or radiation, which is not discussed in the review.

Please include a section on standard treatment of OPSCC, the state and sense of checkpoint inhibitors in relation to standard treatment, in HPV+ OPSCC; how, the E6/E7 therapeutic vaccinnation is expected to be performed in the treatment schedule as first line treatment and adjuvant therapy?

OPSCC can be treated as surgical first line, followed by PORT, or with radiochemotherapy as first line. HPV levels can be followed by circulating tumor DNA measurements or similar. Studies are available that intend to follow up the HPV+ OPSCC treatment schedule and define treatment times, also for E6/E7 vaccines. The only discussion about immune checkpoint regulators targeted therapy is not comparable with the promised de-escalation strategy. 

In 2022 RNA vaccines should be also mentioned and discussed. Please find a possible source for this point in a recent article:

Oncoimmunology. 2019; 8(9): e1629259., DOI: 10.1080/2162402X.2019.1629259

As mentioned in the manuscript, the response rate in OPSCC to standard therapy is relative high. Nevertheless, very few cases do not benefit from the therapy, which require alternative treatments. Do these cases have a chance by the therapeutic vaccines treatments

Response: We appreciate these comments. We now mention more clearly what is the actual standard of care for HPV+ OPSCC (Surgery associated with radiotherapy and/or chemoradiotherapy) and in which directions deintensification strategy go. Based on your suggestions, we expanded the topic 7 and also we included the discussion about RNA vaccines as well, including the recent citation you mentioned.  Thank you.
